# Influenza vaccination hesitancy in large urban centers in South America. Qualitative analysis of confidence, complacency and convenience across risk groups

Miguel Ángel González-Block[1,2,3]*, Blanca Estela Pelcastre-Villafuerte[2], Daniela Riva Knauth[3], Andréa Fachel-Leal[3], Yamila Comes[4], Pedro Crocco[5], Laura Noboa[6], Berenice Rodríguez Zea[7], Mónica Ruoti[8], Sandra Patricia Díaz Portillo[9], Elsa Sarti[10]

1 Universidad Anáhuac, Mexico City, Mexico, 2 Instituto Nacional de Salud Pública, Cuernavaca, Mexico, 3 Universidade Federal do Rio Grande do Sul, Porto Alegre, Brazil, 4 Maestría en Sistemas de Salud y Seguridad social, Universidad ISALUD, Buenos Aires, Argentina, 5 Escuela de Salud Pública de Chile, Santiago, Chile, 6 Facultad de Ciencias Sociales, Universidad de la República, Montevideo, Uruguay, 7 Ministerio de Salud, Lima, Perú, 8 Universidad Iberoamericana, Asunción, Paraguay, 9 Evisys Consulting, Mexico City, México, 10 Sanofi Pasteur LATAM, Mexico City, Mexico

* miguel.gonzalezblock@gmail.com

**Data Availability Statement:** I have uploaded our focus group transcripts in the Dryad repository.

## Abstract

Influenza vaccination coverage in countries of Latin America is low among priority risk groups, ranging from 5 to 75% among older people. This paper aims to describe and analyze the determinants of influenza vaccination hesitancy through the lens of the 3C model of confidence, complacency and convenience among middle-class, urban risk group populations in Brazil, Chile, Paraguay, Peru, Uruguay, countries in South America with contrasting vaccination coverage. Focus groups were conducted among four risk groups: pregnant women, mothers of children aged <6 years, adults with risk factors, and adults aged ≥60 years in samples of urban residents. Adults with risk factors expressed the most detailed perceptions about confidence in the vaccine. A wide range of perceptions regarding complacency were expressed across risk groups and countries, with pregnant women and mothers showing greater concerns while convenience had a narrower and generally more positive range of perceptions. Participants from Chile and Paraguay expressed the most contrasts regarding confidence and complacency. Information and communication strategies need to be tailored for risk groups while confidence and complacency should be addressed in synergy.

## Introduction

Influenza is a respiratory viral disease whose symptoms can be confused with those of a common cold, easily escalating to a more serious condition and even leading to death. Influenza cases are estimated at around 1 billion per year worldwide, of which 3 to 5 million are severe. Modelling studies suggest between 291 243 and 645 832 influenza deaths occur annually [1].

Please see: Gonzalez-Block, Miguel Ángel et al. (2021), Focus group transcripts for Influenza vaccination hesitancy in large urban centers in South America. Qualitative analysis of confidence, complacency and convenience across risk groups, Dryad, Dataset, https://doi.org/10.5061/dryad. r7sqv9scg.

**Funding:** This project was financially supported by Sanofi Pasteur. Elsa Sarti is an employee of Sanofi Pasteur and played a role in the design of the study. All other authors have no relevant conflicts of interest to report.

**Competing interests:** I hereby acknowledge that our project was financially supported by Sanofi Pasteur. This does not alter our adherence to PLOS ONE policies on sharing data and materials. One of our co-authors, Elsa Sarti, is an employee of Sanofi Pasteur. All other authors have no relevant conflicts of interest to report.

H1N1 influenza spread out with particular force in South America, registering over 2.6 million cases in 2009 and particularly affecting Argentina, Brazil and Chile with over 600 thousand estimated cases in each country [2].

According to the World Health Organization, vaccination is the most effective measure to prevent influenza infection [3]. However, vaccine efficacy was low at first, with figures below 60% across age groups and among older adults ranging from 9 to 60%. Vaccine efficacy increased above 60% with the introduction of the high dose vaccine, reducing mortality by up to 80% [4]. Influenza vaccine has been reported with up to 90% of effectiveness in the case of healthy adults, while in older adults, effectiveness can be of 60%, reducing mortality by up to by 80% [5]. Most Latin American countries influenza vaccination policies mandate it for children aged under 6 years, people with chronic comorbidity, adults aged over 60 years, pregnant women within 20 weeks of gestation, or women in the postpartum period [6]. Yet countries in the region have ample opportunity to increase coverage. In the countries comprised by Brazil and those in the Andean Region and the South Cone, average vaccination coverage in 2018 for older adults ranged from 5 to over 75%, with an average of 61.6%. Among children under six years of age, average coverage was 58.4% and 56.7% for pregnant women and 76.7% for adults with chronic conditions [7]. Coverage in Argentina, Bolivia, Brazil, Chile, Colombia and Ecuador is above average when considering all risk groups, while coverage in Paraguay, Peru, Uruguay and Venezuela is below average.

The decision-making process followed by the population to get vaccinated is immersed in a specific social context of beliefs and perceptions as well as considerations of the availability of the vaccine and its costs. Mistrust and doubts about the safety and efficacy of the seasonal influenza vaccine represent one of the main barriers to vaccination in various world regions and constitutes a threat to vaccination programs [8, 9]. Divergence of vaccination uptake across countries in South America attests to the importance of analyzing the role that diverse supply and demand factors play.

The World Health Organization's Strategic Advisory Group of Experts proposed the concept of vaccine hesitancy with the objective of analyzing the social factors that lead either to a delay in the acceptance or to the rejection of vaccines despite availability of vaccination services [10]. Vaccine hesitancy is the result of a complex interaction of behavioral and societal factors whose intervention requires an integral approach. Different conceptual models have been proposed to address the complexity, applicability, and potential usefulness of vaccine hesitancy indicators, as well as for the design of surveys and interventions that can be applied locally and globally [11]. The "Three Cs" model of vaccine hesitancy analyzes three groups of determinants: confidence, complacency, and convenience, and has been considered as one of the most useful models of vaccine hesitancy given that it is intuitive and easy to understand and apply. This model is derived from the sociomedical literature and is complete as well as simple, allowing us to understand the complex vaccination decision-making process [12]. Confidence is the degree of trust in the effectiveness and safety of the vaccine, in the system that delivers the vaccines–including the reliability and competence of the health services and health professionals–, and in the motivations of those who make the decisions to achieve effective access to the vaccines [10]. Lack of confidence is caused by strong negative attitudes towards vaccination, which can be influenced by misinformation about vaccination risks, by affiliation to anti-vaccine groups or through legitimate concerns regarding vaccine safety and efficacy.

Complacency refers to the degree to which people consider vaccination necessary to prevent a vaccine-preventable disease and is the result of a combination of risk perception, knowledge of the disease and of the vaccine and of prejudices relating to side effects and other reactions, and of the need for vaccination. Complacency is influenced by the relative priorities

assigned to health as against other responsibilities and may stem paradoxically from a lowering of risk perception due to immunization program success. Self-efficacy–the self-perceived or real ability of an individual to take action to vaccinate–also influences the degree to which complacency determines hesitancy.

Convenience of the vaccine is defined by availability, affordability, willingness-to-pay, geographical accessibility, ability to understand and accept vaccine-related information (language, culture and health literacy), the appeal of immunization services, and quality of care [13].

This paper aims to describe and analyze the determinants of influenza vaccination hesitancy through the lens of the 3C model across urban risk groups in five Latin American countries with contrasting coverage levels, selecting Brazil and Chile as high coverage countries and Paraguay, Peru and Uruguay as low coverage countries. Countries with different coverage rates were selected to identify the fullest possible range of expressions of confidence, complacency and convenience and to discuss their implications for vaccine uptake.

## Methods

The protocol was reviewed by authorized ethics committees within each of the study countries, as follows: Brazil, Comissão Nacional de Ética em Pesquisa, 05215918.6.0000.5347. Chile: Comité de Ética de Investigación en Seres, Universidad de Chile, Facultad de Medicina, 191–2018. Paraguay: Comité de ética en Investigación, Laboratorio Central de Salud Pública, 106/2019. Peru: Comité de Ética de Investigación Prisma, CE1651.18. Uruguay: Comité de Ética en Investigación, Instituto Nacional de Salud Pública, 1580.

The study design is multi-center and qualitative, contrasting knowledge, attitudes and practices through focus group technique with homogeneous members of four risk groups. Focus group methodology was chosen over other techniques to address the qualitative aspects of vaccine hesitancy given its capacity to obtain a greater amount of information in a shorter time elicited both through directed queues and the interaction between participants [14]. Furthermore, focus groups would enable us to obtain a first repertoire of perceptions on the basis of which to later address the hesitancy through more structured instruments. Interaction between persons sharing risk group characteristics would elicit the widest possible range of views based on knowledge, beliefs and practices regarding influenza vaccine confidence, complacency and convenience.

Focus groups were evenly distributed in one or two large cities within each country as follows Brazil: Porto Alegre (in the state of Rio Grande do Sul); Chile: Santiago and Valparaíso; Perú: Arequipa and Lima; Paraguay: Asunción and Ciudad del Este, and Uruguay: Montevideo and Salto (Table 1). Participants across the four risk groups lived in lower-middle class neighborhoods. A representative survey of health facility clients in the same neighborhoods undertaken as a part of the same research project to explore quantitative determinants of vaccine hesitancy showed that between 26.9 and 38.8% of elderly adults and adults with risk factors, respectively, had up to primary education, as against pregnant women and mothers of children, among whom only between 8.1 and 9.3% had such lower education levels. Participants from Paraguay and Uruguay tended to have lower education levels across risk groups when compared to participants from the other countries [15].

Focus groups were piloted and initiated in Peru in October 2018 and followed up in the four other countries between March and July of 2019, as soon as ethics committee authorizations allowed, and aiming to hold them as close as possible to influenza immunization campaigns. Table 1 describes the focus group recruitment strategy and characteristics. Individuals

**Table 1. Focus group participants by country and risk group.**

| Country and date | No. of focus groups by recruitment strategy (and risk group) | Place where focus group was held | No of participants by group | | No. of females-males /Average age | Average duration in minutes | Compensation |
|---|---|---|---|---|---|---|---|
| Brazil, June 2019 | 4 Pub | Health services' meeting room | Total 33 | | | 90 | None |
| | | | OA 9 | | 8-1/ 71 | | |
| | | | ARF 8 | | 6-2/ 71 | | |
| | | | PW 7 | | 7/ 35 | | |
| | | | MC 9 | | 9/ 34 | | |
| Chile, May 2019 | 4 OS | Researchers' meeting room | Total 30 | | | 85 | USD 20 gift card |
| | | | OA 8 | | 4-4/ 69 | | |
| | | | ARF 8 | | 5-3/ 47 | | |
| | | | PW 6 | | 6/ 27 | | |
| | | | MC 8 | | 8/ 33 | | |
| Paraguay, March 2019 | 1 Pub (MC) | Pub: Within health services; Priv: Hotel meeting room & researchers' meeting room | Total 29 | | | 90 | Umbrella and toiletry bag' |
| | 3 Priv (OA, ARF, PW) | | OA 7 | | 4-3/ 68 | | |
| | | | ARF 9 | | 8-1/ 50 | | |
| | | | PW 7 | | 7/ 26 | | |
| | | | MC 6 | | 6/ 32 | | |
| Peru, November 2018 | 3 Pub (OA, ARF, PW) | Pub: open air meeting place in park near health services except for PW, held in meeting room. | Total 25 | | | 90 | Diapers only for PW |
| | 1 Priv (MC) | | OA 7 | | 6-1/ 71 | | |
| | | | ARF 6 | | 4-2/ 42 | | |
| | | | PW 6 | | 6/ 27 | | |
| | | Priv: health services' meeting room | MC 6 | | 6/ 35 | | |
| Uruguay, July 2019 | 4 OS | Researchers' meeting room | Total 33 | | | 90 | USD 27 gift card |
| | | | OA 7 | | 4-3/ 68 | | |
| | | | ARF 9 | | 8-1/ 49 | | |
| | | | PW 7 | | 7/ 31 | | |
| | | | MC 10 | | 10/ 35 | | |

Pub: Public services; Priv: Private services; OS: Other Strategy (Facebook call or researchers' contact network); OA Older adults; ARF: Adults with Risk Factors; PW: Pregnant Women; MC: Mothers of children.

with homogeneous characteristics were recruited based on having the attributes of each of the following risk-groups: adults above 60 years of age (OA), adults with risk factors (ARF), pregnant women (PW) and mothers of children under 6 years of age (MC). Focus groups had between 6 and 10 participants with an average of 8, recruited upon leaving health facilities in the case of Brazil, Paraguay and Peru. In Chile, participants were recruited based on a sample frame developed for research purposes and registering risk group attributes and socioeconomic characteristics. In Uruguay, recruitment was based on a Facebook call, filtering for risk group and health service use characteristics. Heterogeneity within focus groups was sought with respect to the vaccination decision (accepted and declined) in order to understand in-depth, the reasons and influences in both cases, as well as the obstacles and facilitators that led to their choice. Focus groups were undertaken in Spanish or Portuguese for the case of Brazil. All participants spoke Spanish or Portuguese and no translation was required. Focus groups were moderated in all cases by an expert focus group researcher, supported by an observer. Focus groups were held in meeting rooms or open spaces within or next to health facilities in

Brazil, Paraguay (one focus group) and Peru, and in research meeting rooms or a hotel meeting room in Chile, Paraguay (three focus groups) and Uruguay.

A discussion guide was developed and organized into knowledge, attitudes and practices sections including each the discussion of vaccine confidence, influenza complacency and vaccine convenience. (The guide is available as a S1 File). Within each section. Focus group sessions were audio-recorded after verbal informed consent was obtained from participants and information was transcribed verbatim. A first code list was developed by two of the authors (BP and MAGB) based on the theoretical framework and the pilot study focus group data from Peru. A coding manual was provided to researchers within countries (YC, PC, AFL, LB, DRK, BR and MR). Codes were inductively refined and detailed by country-level researchers based on focus group data. Detailed codes were reviewed by BP to arrive to a final set of codes for each country. Country data codification was reviewed by BP, DRK, AFL and disagreements involving the allocation of statement to the categories of confidence or complacency were resolved by consensus. Information was encoded with the support of the Atlas-ti v.8 software. Qualitative content analysis was carried out on thematic units of analysis based on the 3 Cs model categories by each of the country researchers. The main focus of analysis were risk groups, followed by cross-risk group analysis. Country case studies were produced by country-level researchers and reviewed by BP and MAGB. Final interpretation was based on comparing country case studies.

## Results

Tables 2–4 present an overview of the range and contrast in perceptions regarding confidence, complacency and convenience by risk group.

### Confidence: Effectiveness of the vaccine

The group of older adults and, to an extent, adults with risk factors showed a wide range of positive evaluations on the effectiveness of vaccines, including the recognition of the importance that vaccines be effective and a more nuanced perception of vaccine potential and limitations. As shown in Table 2, perceptions range from a clear sense of vaccine effectiveness, including limitations, lack of reliability, and a perception of trust or distrust in the context of negative experiences:

> I always got the flu, I felt really terrible. My relatives would call me and say "but it's not possible that you have the flu again. Take vitamin C" After I started taking the vaccine, I never got worried about the flu again, nor even of a cold.
>
> (OA, Brazil).

> I have been vaccinated and [yet] I caught a cold, but [it was] milder (. . .) but I still trusted the vaccine.
>
> (OA, Chile).

> I don't know for sure, but I've heard that in the USA nobody takes the vaccine, because they say that the vaccine is to make more people have the flu.
>
> (ARF, Uruguay)

In spite of perceiving influenza as a grave threat, the appreciation of effectiveness is ambivalent across risk groups, as attested by mothers:

**Table 2. Thematic analysis of confidence by risk group across the five countries.**

| Category/ Group | Older adults | Adults with risk factors | Pregnant women | Mothers of children |
|---|---|---|---|---|
| **Effectiveness of vaccine** | • Vaccine is trusted and accepted as key preventive action<br>• Vaccine known to be effective<br>• Compliance during campaigns associated with trust in the vaccine<br>• May accept vaccination in spite doubts about efficacy<br>• Influenza is contracted even after vaccination | • Clear concept of vaccine efficacy<br>• All vaccines are only partially effective<br>• Vaccine effectiveness widely shared<br>• Vaccine is not reliable, inefficacious<br>• Doubts on efficacy<br>• Trust clearly associated with positive experiences<br>• Some trust in spite of negative experiences<br>• Feel protected by vaccine<br>• Distrust when negative experiences perceived | • Few express trust<br>• Benefit of vaccine is prevention of influenza<br>• Question vaccine efficacy, but no serious side-effects<br>• Adequate, timely information leads to trust, even when there are side effects | • Vaccination experience related to perceptions of efficacy<br>• Vaccine is controversial<br>• Doubts on range of influenza strain protection<br>• The more specific information is available, the more trust |
| **Safety of vaccine** | • Vaccine can weaken capacity to fight influenza<br>• Vaccine known to be safe<br>• Vaccine associated with allergies and reactions<br>• Fear of getting sick leads to reject vaccination<br>• May accept vaccination in spite doubts about safety<br>• Misinformation can blur safety concerns<br>• Low information related to low trust | • The vaccine can cause influenza<br>• Vaccine safety widely shared<br>• Doubts on safety<br>• Vaccine rejected to avoid mild side effects: Fever and cold<br>• Vaccination during a respiratory episode can be harmful<br>• Fear of needles<br>• Fear that the vaccine is expired<br>• Information demanded on adverse side effects and on their impact on comorbidities | • Do not have a clear understanding of disease<br>• Vaccine rejected out of fears of health consequences for baby<br>• Discomfort and pain typical of virus inoculation<br>• Vaccine perceived as not natural and potentially harmful | • Vaccine can cause the disease<br>• Vaccination experience related to perceptions of safety<br>• Little knowledge about influenza and the vaccine<br>• Mercury in the vaccine perceived as autism risk<br>• Information on influenza and the vaccine demanded, as specific as possible<br>• Lack of information during vaccination campaigns<br>• Face-to-face to information at health centres is valued |
| **Trust in health system** | Doubts due to varying vaccine and disease names<br>• Vaccine perceived as reliable, especially if administered in health facilities<br>• Yet institutions can be mistrusted<br>• Information mostly sought from health providers | • Annual vaccination can prevent influenza<br>• Distrust related to insistence on repeating vaccination annually<br>• Trust associated with medical advice and to trust in health institutions<br>• Higher quality of vaccine in the private sector<br>• Knowledge given by physicians is trusted<br>• Satisfaction with information available<br>• Distrust due to opaque interests of vaccine producers<br>• Main sources of information are health providers and during campaigns | • Trust as an "act of faith" in health system<br>• Doubts due to varying vaccine and disease names<br>• Confidence in medical referrals and health providers<br>• Question the need for vaccination, even if prescribed<br>• Health team is main source of information | • Strong adherence to child vaccination schedules<br>• Journalists on TV and the press introduced fears of vaccination<br>• Health team and personal experience are more important sources of information than campaigns |

*I don't trust it entirely. People say that if you have the flu you can die... then there are those who take it [the vaccine] and you can still end up really sick. Then there are those who don't take it [the vaccine] and who don't get sick.*

*(MC, Brazil).*

**Table 3. Thematic analysis of complacency by risk group across the five countries.**

| Category/Group | Older adults | Adults with risk factors | Pregnant women | Mother under 5 years old |
|---|---|---|---|---|
| **Perceptions of influenza risk and severity** | • Influenza is a "a more serious flu", easily transmitted, with more intense symptoms, which can result in complications and death<br>• No clear definition of influenza; confusion with dengue<br>• Influenza is serious and potentially mortal but treatable; needs care and medication<br>• Perception of risk severity related to experience of contagion by self or close contacts<br>• Severity also perceived from mass media<br>• All the population has same risk of contracting influenza<br>• Everyone can get sick, but older adults get worse<br>• Elderly are special risk group<br>• Grandchildren more at risk<br>• Influenza is vaccine preventable<br>• Not a treatable disease; it has to be overcome<br>• Importance of timely medical care | • Influenza perceived as "more intense" flu varying according to person's state of health<br>• Influenza is a dangerous, potentially mortal disease<br>• Influenza can lead to complications of pre-existing illness<br>• "Low defenses" favor influenza<br>• Relationship with pre-existing risks not always recognized | • Influenza is a serious and potentially mortal disease<br>• Influenza symptoms similar, but stronger, than those of common cold or flu<br>• Influenza is a complication of respiratory diseases<br>• Ignorance of disease symptoms<br>• Confusion with dengue and swine flu<br>• Influenza not a serious disease<br>• Highly infectious among the general population<br>• Pregnancy not a risk for influenza<br>• Older adults and children more at risk<br>• A preventable disease<br>• The vaccine lasts for several years<br>• Can be cured by health providers | • Influenza is a normal disease in childhood<br>• Influenza is a "strong flu"<br>• A severe, potentially mortal disease<br>• Severity is learned from close contacts<br>• Influenza symptoms are similar to flu, but stronger<br>• Influenza is a complication of a respiratory process<br>• Influenza caused by climate change and exposure to cold<br>• Everyone is equally exposed<br>• Children more exposed due to behavior<br>• People more at risk are those suffering other diseases, pregnant women and older adults<br>• A treatable disease by doctors<br>• Influenza is vaccine preventable, although some doubt it |
| **Alternatives for prevention** | • Multiple prevention methods<br>• Vaccine is the socially validated main prevention method<br>• Principal prevention methods are hygiene, diet and home remedies<br>• Fear of injections an important reason to reject vaccine<br>• Hygiene and health foods with vitamin C can prevent influenza | • Multiple prevention methods, but vaccine is best<br>• Hand washing and wearing face masks more important than the vaccine<br>• Vaccination is the main form of prevention when influenza is perceived as important<br>• Healthy habits are complementary to vaccination Prevention more important than treatment<br>• A healthy person better avoid the vaccine and its secondary effects | • Existence of the vaccine is well known<br>• Vaccine is one among other preventive measures<br>• Healthy pregnancy does not require vaccination<br>• Influenza can be prevented hygiene and a good diet with vitamin C<br>• Avoiding contact with sick is important (sharing utensils)<br>• Vaccine not demanded if not prescribed by doctor | • Vaccine is one among other preventive measures<br>• Vaccinate avoided if hygienic-dietetic measures practiced<br>• Vaccine complementary to healthy diet |
| **Self-efficacy** | • Important to know cases of diseases and how to prevent it<br>• Demand information on severity<br>• Information on adverse side effects is demanded | • Demand information for early detection<br>• Information about the disease, vaccine and its side effects is demanded | • Vaccine is important given medicine contraindications during pregnancy<br>• Information about the disease, disease prevention and the vaccine is demanded<br>• Knowledge demanded on the impact of vaccine on baby | • More specific information on influenza is demanded (characteristics, strains covered)<br>• Keeping vaccination schedule/passbook up to date is an incentive |

Mothers more than other groups expressed the need for specific information on vaccine effectiveness to enhance their trust:

> *I think more information is needed. I do not get carried away by people, of what they are saying. I think you have to trust, and you have to get informed, that the doctor explains [the*

**Table 4. Thematic analysis of convenience by risk group across the five countries.**

| Category/Group | Older adults | Adults with risk factors | Pregnant women | Mother under 5 years old |
|---|---|---|---|---|
| **Availability and geographical accessibility** | • High availability of vaccine<br>• Vaccination more frequent during campaigns<br>• Easy access to health centers, both public and private<br>• Health facilities and mobile posts provide vaccine during campaign<br>• Good proximity<br>• Options at municipal level<br>• All posts open regardless of health coverage<br>• Not known which services treat influenza<br>• Vaccine can be bought in pharmacy for home application<br>• Vaccines are always available at public and private health centers | • Easy access to health centers for vaccination, both public and private<br>• Ease of being vaccinated in campaigns<br>• Vaccine not always available<br>• Diversity of vaccination centers (hospitals, polyclinics, state institutions, mobile)<br>• Know which health services treat the disease<br>• Persons outside priority risk groups have difficulties in accessing<br>• Neighborhood councils offer vaccination<br>• Health facilities nearby<br>• All posts open regardless of health coverage<br>• Vaccine can be bought in pharmacy for home application<br>• Vaccination only during campaign<br>• Medical care received for pre-existing diseases facilitates access to influenza information | • Easy access to health centers for vaccination, both public and private<br>• Ease of access, even after campaign ends<br>• Vaccine not always available<br>• Vaccine availability in workplaces<br>• Pregnancy facilitates vaccination<br>• All posts open regardless of health coverage during the campaign<br>• Good access to public and private health centers in metropolitan area<br>• Closeness to health facilities<br>• Vaccine applied in prenatal care<br>• Vaccine can be bought in pharmacy for home application<br>• Vaccination in the campaign<br>• Do not know where and when to get vaccinated | • Difficult access to health facilities due to closures<br>• Lack of vaccines in some health facilities<br>• Easy access to health centers for vaccination, both public and private<br>• Know where to vaccinate children<br>• Educational institutions and mobile vaccines units identified<br>• Vaccine only available during campaigns<br>• No consensus on the availability of the vaccine<br>• Good access to public and private health centers in metropolitan area<br>• Municipal educational establishments vaccinate children<br>• Easy access to vaccination posts<br>• Closeness to health facilities<br>• Vaccine can be bought in pharmacy for home application<br>• Difficult to travel with children<br>• Vaccination sites have been reduced |
| **Affordability and willingness-to-pay** | • Free public health system used<br>• Vaccines can be expensive in private services | • Free access | • Free access<br>• Those who were vaccinated outside pregnancy had to pay | • Free access |
| **Understanding and appeal** | • Self-perceived as a privileged group in campaigns<br>• Insufficient information<br>• Insufficient information on campaigns<br>• Feel a strong link with the health service<br>• Feel satisfied with knowledge on vaccine and the flu | • Everyone knows where to get vaccinated<br>• They are well linked to health services to monitor chronic conditions | • Demand more information on locations and opening hours | • Lack of information about campaigns<br>• Do not participate in vaccination campaigns<br>• Media and close contacts are sources of information<br>• Information on prevention from TV and social networks<br>• Reliable information on vaccine demanded<br>• Adequate vaccine knowledge available on importance of campaigns |

*(Continued)*

**Table 4.** (Continued)

| Category/Group | Older adults | Adults with risk factors | Pregnant women | Mother under 5 years old |
|---|---|---|---|---|
| **Quality of service & cultural appropriateness** | • Vaccination is fast<br>• Main source of information are health providers<br>• Positive evaluation of vaccination at public and private health centers<br>• Attention is prompt and friendly<br>• Vaccination is often late<br>• Conflict between vaccination and cultural perceptions of state of being (the vaccine must be applied without being "cold")<br>• Vaccination information comes from health centers, mass media (vaccination campaigns).<br>• Excellent treatment from service providers<br>• Good treatment in health care in services and campaigns | • Positive evaluation of vaccination at public and private health centers<br>• Attention is prompt and friendly<br>• Vaccination information comes from health centers, mass media (vaccination campaigns).<br>• Satisfaction with the treatment and prompt attention | • Positive evaluation of vaccination at public and private health centers<br>• Attention is prompt and friendly<br>• Vaccination information comes from health centers and campaigns<br>• Good treatment at the vaccination center<br>• Prenatal check-ups as main source of information | • Positive evaluation of vaccination at public and private health centers<br>• Attention is prompt and friendly<br>• Vaccination information comes from health centers<br>• Barriers to treatment<br>• Personnel need training<br>• Cultural barriers.<br>• Bad treatment in some establishments<br>• Rapid service |

*importance of the vaccine] to us; and communication and contact with health personnel are important.*

*(MC, Peru).*

## Confidence: Safety

All groups expressed, paradoxically, that the vaccine is safe and that it has risks of its own. Except pregnant women, informants of all risk groups expressed concerns that the vaccine may be ineffective and cause adverse reactions:

*I was vaccinated the first year that the influenza vaccine came out, but that year was the worst in my life, . . . it gave me a fever, then for the following year, my daughter-in-law. . . said "we are not going [to vaccinate] you". . . I will never get that vaccine again. . . apparently it does not suit me.*

*(OA, Paraguay).*

Aspects such as fear of needles, the adduced presence of mercury and its relations to autism, and ignorance on side effects the vaccine can have on babies contribute to raising doubts about safety. One mother from Chile with experience of pre-term births expressed having to be particularly careful in case contaminated vaccines cause autism:

*In the last resort, I prefer that [vaccines] be mercury-free. . . because I know of cases of people with autism who associate it with that.*

*(MC, Chile).*

Mothers and pregnant women also expressed concern with potential side effects from the vaccine:

*I think that the vaccination campaigns. . . on television, are informative but superficial. [They should be] more in-depth [stating] side effects, [how to] differentiate the flu from a common cold, at what point to get the vaccine if you have a cold. The information on TV is about how to get the vaccine, get vaccinated, but why, or for what benefits, that I think is lacking.*

*(MC, Chile).*

## Confidence: Trust in the system

Confidence in the information offered by service providers varies across risk groups. Confusion with the wide range in vaccine nomenclature i.e. "H1N1", "influenza", "swine flu" was widely shared. All groups also expressed the need to have more, and more specific, information on influenza and on the vaccine.

*It's like a mutation from the common flu. It was called the swine flu. I do not know very well how it started and each time it comes back, it is as if it continues to mutate. There's the H1N1 and I think there are other letters, there are several. Actually, that was the one that affected the most.*

*(MC, Uruguay).*

Adults with risk factors expressed the most nuanced notions of trust (or distrust) with the health system, both as a whole and with vaccination services and campaigns specifically. The fact that influenza vaccination schedule needs to be updated every year engenders mistrust regarding the vaccine's effectiveness.

*My boss works in health and says that every year the strains change, yet when the vaccine comes here it is [with] the old strain.*

*(OA, Uruguay).*

Trust in public or government health services varied across countries, but in all groups, there was trust in general. Public services are seen as more reliable because they have historical expertise and know-how, besides having greater resources than the private sector:

*I've heard that the vaccine cold chain works better in the public health services than. . . in the private services.*

*(ARF, Paraguay)*

Some participants expressed that the vaccine in private health services might be of a better quality than in their public counterparts, while also having shorter waiting times.

Health providers were by far the most common source of trusted information on vaccination across risk groups.

*When the doctors explain things to me, I understand them better, and they told me: get your shot for the flu, or else it will be very bad. And so that was it, I went and got the vaccine. Really, that was the main reason I got the vaccine.*

*(MC, Uruguay).*

*There are doctors who. . . tell you where you should go to get a vaccine–and they follow up on you, they ask you if you've got your shot. This is why I believe this is the best vaccine, I feel confident, I am sure that vaccines are good for everybody.*

*(ARF, Paraguay)*

Despite being a trusted source of information, medical advice regarding the vaccine is not always followed through. Reasons for not getting the vaccine in these cases are not due to a lack of confidence and may be associated with factors regarding convenience and complacency.

Another concern expressed with trust in the system was that vaccination program priorities could be driven by pharmaceutical industry interests. In an extreme, some participants suspected the pharmaceutical industry had financial gains with the process of updating influenza strains. The commercial interest related to the vaccination programs raises doubts the credibility of vaccination program guidelines and of public health professionals' advice:

*I think [vaccination programs have a] pharmaceutical industry issue. I have a certain conspiracy theory, but then it's something of mine. . . I read about it. . . I also think that the pharmaceutical industry promotes so many tests–they test, test, test. It seems to me that [virus strains] could be something that laboratories develops.*

*(PW, Brazil).*

*It is a business of the laboratories that treat us like guinea pigs.*

*(OA, Uruguay).*

Comparing the perception of confidence across countries, more often questioning of vaccine effectiveness was expressed in Paraguay than in other countries across risk groups. In this country participants reported commonly the use of traditional medicine and herbs in the prevention of influenza and for the treatment of disease symptoms, a qualitative association that suggests that trust in traditional medicine could rival trust in the vaccine Participants from Brazil, Paraguay, Peru and Uruguay questioned vaccine effectiveness only in the case of some risk groups, especially mothers of young children.

Participants from all risk groups in Chile expressed positive views about vaccine safety. In Paraguay, by contrast, the idea that the vaccine can cause the disease was widespread. In Peru and Brazil, mistrust concerning the safety of the vaccine occurs more among adults with comorbidities, while in Brazil questions arose about the safety of the vaccine during pregnancy. Greatest distrust in the health system was expressed in Paraguay, especially regarding the view that the vaccine is, in general, offered late. Related to this, participants in this country expressed more trust in traditional medicine and in pharmacies than participants from other countries. In Uruguay and Brazil, participants in some risk groups, particularly adults with comorbidities and pregnant women, say that doctors have different positions concerning vaccine recommendation, raising doubts about the safety of the vaccine.

## Complacency: Influenza risk and severity

Influenza was perceived by most risk groups as a serious illness, aggravated by the fact that there are no pharmaceutical cures and that it may be life-threatening (Table 3).

*[Influenza is of] high risk, a person can die. My niece got sick last year and badly, [she had] seizures, high fever. [She was given] very expensive antibiotics and was hospitalized.*

*(ARF, Chile).*

*And curing the flu, it does not cure; it has its process, you can alleviate the symptoms with all that is sold in the market.*

*(OA, Uruguay).*

Pregnant women were most ambiguous in discussing the risks regarding the vaccine and the disease. Some participants argued that it would be best to take the vaccine, because if they had the flu they would have limited options due to drug contraindications during pregnancy:

*Not being pregnant, I can take anything. . . But now I know I can't. I think that's why [I got vaccinated], I don't know if I'm going to give it to myself next year. [Pregnancy] is like a stage where I feel more vulnerable in health.*

*(PW, Uruguay).*

*It seems that [influenza] is stronger in pregnancy, they told me that I had to be vaccinated. . . since I cannot take anything.*

*(PW, Chile).*

Other pregnant women perceived influenza with ambivalence with respect to its severity:

*I think influenza can be easily cured. . . I don't know anyone close to me with a flu that knocks you down. But from everything I've read about it, influenza can be very complicated, there are cases of death. But in my family environment and in my work, I work in a salon, it is not very big and there has never been a case of someone who had to be bedridden.*

*(PW, Brazil)*

Confusion between influenza and other respiratory diseases led also to complacency, where influenza is seen as a cold (*resfriado*) with more intense symptoms or as a complication of a common cold:

*Influenza is like a kind of flu [gripa], a variety of flu so to speak that it can be contracted [if the person has] low defenses or is poorly cured of the flu.*

*(PW, Peru).*

*It is a flu disease that, when not cared for, can become pneumonia, it has more risks and it is a little difficult to cure and that is why you have to be vaccinated every year.*

*(ARF, Peru).*

*Let them tell us the specific benefits of the vaccine; we do not know in writing what the benefits are [for children]. Not catching the flu, but what else? Maybe it is also useful for respiratory infections, to make them milder, I don't know, anyway. [We need] something that one can read, short and clear, not technical information.*

*(MC, Uruguay).*

Individuals did not perceive themselves to be in a special risk category, except in a few instances with pregnant women–as seen above–and with older adults. Rather, everyone is deemed equally exposed to contracting influenza. Among mothers, what makes children vulnerable to catching influenza are certain behaviors, such as walking barefoot or going out without proper clothing:

> *It's impossible to avoid [the flu] with children. . . children will be children*! *They go about in their bare feet, they drink ice cold water, they don't want to put on proper clothes. Not even socks, even when its winter*!
>
> *(MC, Brazil).*

## Complacency: Alternatives for prevention and treatment

All risk groups expressed that there are multiple ways of preventing influenza besides the vaccine, although the vaccine was perceived as the most important, clearly so by mothers and pregnant women (Table 4).

> *I have many measures that I had to follow with my son [such as] when the air is very bad not taking him to public places. . . But the vaccine is the most important, because. . . [children] at school don't remember to keep washing their hands, and there are many bugs there, but the vaccine is always [effective].*
>
> *(MC, Chile).*

> *Influenza can be prevented, if you are careful enough with what you take, taking care not to infect others and [that they] not to infect you, of course you need vaccinations and that is a way of prevention.*
>
> *(PW, Peru).*

Healthy food and hygiene are commonly mentioned as complementary to vaccination, but also as substitutes:

> *I think that obviously a healthy diet with vitamin C is undeniable that helps us to be prepared. . . I can prevent it staying in my house, not being in crowded environments, especially washing my hands after riding buses, taxis, supermarkets. I think you can warn others if you are sick, not receive visits, not visit anyone.*
>
> *(OA, Uruguay).*

> *I don't much like vaccines in themselves. . . Because I feel that they are not a very natural thing, the body itself has its defenses and one can protect oneself through food.*
>
> *(PW, Chile)*

Elderly adults and mothers expressed that you could mitigate some influenza symptoms; this, in turn, may moderate risk perception and induce alternative preventive actions.

> *A home remedy, a good lemon tea with honey and a splash of alcohol, so that the body perspires, relieves itchy throat, a couple of anti-flu [medications] and do not go out to the cold.*
>
> *(OA, Uruguay).*

Comparing the perception of complacency with influenza across countries, adults with risk factors in Uruguay perceived themselves as particularly vulnerable. With regard to pregnant women, in Brazil, Paraguay and Uruguay they did not see themselves as particularly vulnerable to the flu, as against Chile and Peru where vulnerability was clearly expressed. Participants across risk groups from Paraguay believed influenza can be cured, a perception that could compete with that of vaccines as an alternative to address the disease. Participants from Paraguay and Uruguay privileged behavioral measures for prevention of influenza such as hygiene and diet as well as home or traditional remedies. In Uruguay, pregnant women were more willing to take the vaccine because they anticipate they will have limited resources if they eventually fall ill, considering restrictions due to their pregnancy.

## Convenience: Availability and geographical accessibility

All risk groups perceived relatively good access to influenza vaccination in general. The vaccine is referred as available within health facilities during campaigns.

*You can get [the shot] anywhere, when they launch the national campaign. . . You have to look for the schedules and see when its best for you. It is easy to access.*

*(MC, Uruguay).*

In Brazil there was consensus among all risk groups regarding availability and ease of access in public health services with the health system, the Sistema Único de Saúde (SUS). However, mothers complained of closures of those facilities outside the health system that no longer offered the vaccines as well as changes in primary health daily operations:

*Schools and day care centers used to [provide the vaccine,] not anymore. . . It used to be that in the past there were community agents who went to the houses. Now you have to go to the clinic to give the children a vaccine.*

*(MC, Brazil).*

The campaigns are limited in their duration and target those most vulnerable to influenza. This in turn results in difficulties when the population seek the vaccine outside of the campaign period.

*I went to the vaccination post and I did not find the vaccine. . . The vaccine should be given all year round [in public facilities], because otherwise the ones that commercialize the vaccines are the clinics, the private [businesses].*

*(ARF, Peru).*

Even during the campaign, there were some complaints of stock-outs, but these were not shared by most participants.

*[The vaccine] ends quickly and you need time to go; most [of us] work, to get a leave is very complicated because you have to estimate [when] you will have to go again.*

*(MC, Paraguay).*

*The mutualists [health providers] ran out of doses very quickly. This time I got it at a poly-clinic at the municipality, which was also [open] in specific days.*

*(PW, Uruguay).*

The campaign duration seems to be too short for many who want to use these health services. There were complaints regarding vaccine services scheduling restrictions, especially for participants who work throughout the day and find it difficult to attend health services.

In Brazil expressions were prominent regarding access restrictions for specific risk groups, and the putting in place of stratagems to gain access and avoid payment of private providers:

*You have to bring your child, then on that day you pay for public transport. . . Then, you have to pay [again if you did not find it the first time] for public transport for one, for two. . . That's why it is good to [make the vaccine available] to everyone.*

*(OA, Brazil).*

*I lied that I had asthma and bronchitis to be able to get [the vaccine] at the clinic, because otherwise I would have to pay. And we are not able to pay for a vaccine.*

*(PW, Brazil).*

Confusion was manifested between seasonal and on-demand vaccination schedules:

*This flu vaccine is. . . it only on campaign, on campaign day, or at the clinic? Because I think it should be not only [available] on campaign days. . . in the children's [vaccination] card there are all doses, but there should also be H1N1 as well.*

*(MC, Brazil).*

Focus group participants across all countries highlighted that there are no problems regarding access to the influenza vaccine during campaign periods and for priority risk groups. They know where to get the vaccine and, in general, health services are close by and easily accessible.

## Convenience: Affordability and willingness-to-pay

The vaccine was generally perceived as free at the point of delivery in government facilities during the campaigns, but expensive in private facilities:

*The vaccine for those who are [cared for by] private [providers] is expensive. . . More than 100 reais (U$25), [or even] 180 reais (U$45). So many people don't do it due to the lack of funds.*

*(OA Brazil).*

*They did not charge me because it was in the campaign, so it was free at the public health services . . . I always go during the campaigns, because it is free.*

*(OA, Peru)*

Influenza vaccine was generally perceived as accessible from the economic perspective in all countries, with participants reporting that it could be obtained free of charge within public health facilities.

## Convenience: Understanding and appeal

All groups except adults with risk factors complained of a lack of information on the vaccine and its availability.

*There is no certain information that in such a place they can go whenever they want [to be vaccinated].*

*(PW, Uruguay).*

The little vaccine information is mostly reported to be received across risk groups from vaccination campaigns and health posts.

*I have a health service five minutes from my house and they usually go out to campaign, they inform you, they leave you the brochures and they. . . give you the vaccine and everything.*

*(PW, Peru).*

*In the public health services, I got my shot quickly. This last time I went to a clinic of the municipal government, it was in specific days of the week. Maybe they offer the vaccine in other places, but there really isn't any clear information about what places to go or when we can go.*

*(PW, Uruguay)*

In Brazil older adults and pregnant women recognized themselves as being privileged groups. Older adults are treated as campaign priority and are stimulated by health providers to get their vaccine in their routine consultations. The subject of vaccines is also brought up by health professionals during prenatal care.

*I always take the influenza vaccine, every year I get my shot. . . it's just wonderful. I leave the clinic happy, because I already got my shot. I feel really great.*

*(OA, Brazil)*

In some cases, telephone reminders were mentioned as facilitators. Access to insurance was also a facilitator of convenience:

*My mom gets [the influenza vaccine] every year in the doctor's office; they call her on the phone and tell her that she has to get vaccinated on that day.*

*(ARF, Chile).*

*As I have insurance, they know that I am lacking the vaccine and they simply tell me 'sir, you need to be vaccinated for influenza' and they. . . vaccinate me.*

*(OA, Peru).*

Comparing the convenience of the influenza vaccine across countries, participants from Paraguay expressed least knowledge about influenza and the vaccine. Participants from Chile, on the other hand, expressed a higher level of knowledge. In other countries, participants expressed the need the have more information on the disease and on the vaccine.

## Discussion

To our knowledge, this paper analyzes for the first time the range of knowledge, attitudes and practices expressed with respect to influenza vaccine hesitancy by members of high-risk groups living in large urban areas of countries in South America. In Latin America the only previous qualitative studies on influenza vaccine hesitancy focused on Peru [16], while studies in Brazil have focused on pediatric vaccine hesitancy [17].

Our results suggest that confidence is the most important construct associated with influenza vaccine hesitancy across the four risk groups, mostly through perceptions of mistrust in vaccine safety and effectiveness and, to a lesser extent, mistrust in the health system (Fig 1). Moreover, some groups expressed the fear that the vaccine itself might pose risks, having side effects, specially to those who were thought to be in a vulnerable condition, such as pregnant women and adults with risk factors, a perception that was particularly acute in Brazil. Our study suggests that confidence in the vaccine in the five countries studied and across risk groups was higher in Chile and lower in Paraguay, with marked contrasts regarding vaccine effectiveness, safety, and trust in the health system. Both our qualitative and quantitative results are congruent with the national influenza vaccination rates observed across countries, where

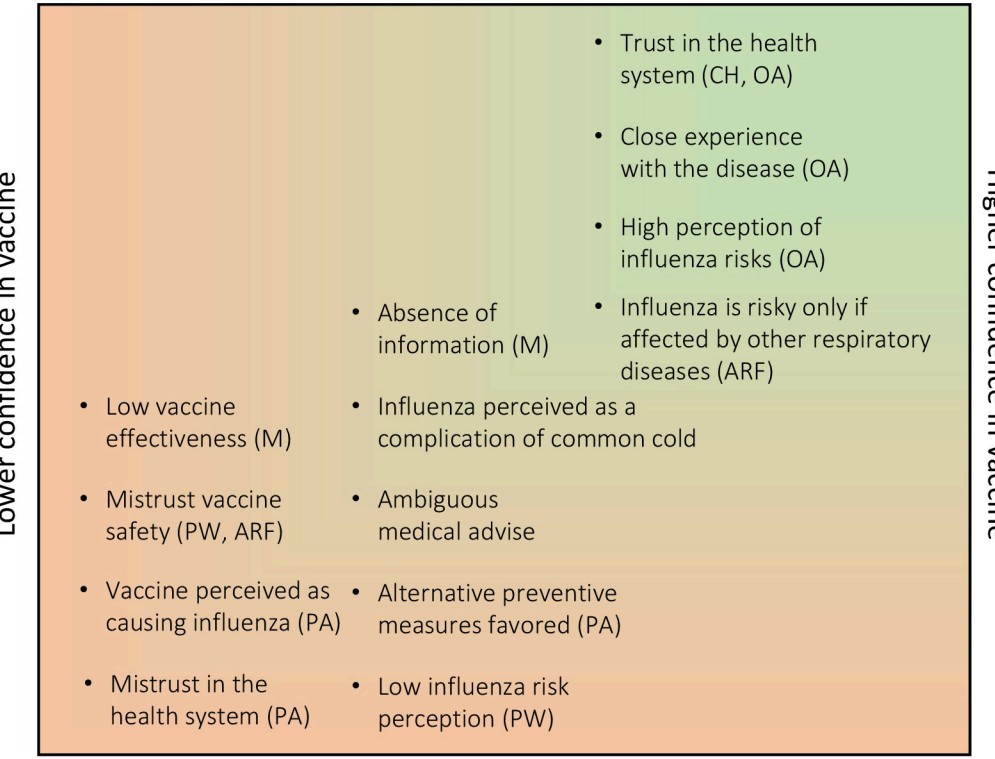

**Fig 1. Most important influenza complacency and vaccine confidence issues across risk groups and countries.** Confidence and complacency may be reinforcing and lead to a positive situation of high confidence and low complacency as clearly observed among older adults and across risk groups in Chile. A less positive situation can be observed among adults with risk factors, followed by mothers of infants and pregnant women, and by across risk groups in Paraguay. Risk groups: ARF Adults with risk factors, MC Mothers of children under 6 years of age, OA Older adults, PW Pregnant women. Countries shown: CH Chile, PA Paraguay.

**Table 5. Influenza vaccination coverage by risk groups in selected countries of South America.** 2018 or most recent year.

| Risk group | Brazil | | Chile | | Paraguay | | Peru | | Uruguay | |
|---|---|---|---|---|---|---|---|---|---|---|
| | **Range** | **% coverage** | **Range** | **% coverage** | **Range** | **% coverage** | **Range** | **% coverage** | **Range** | **% coverage** |
| **Children** | 6M to 6Y | 88 | 6M to 5Y | 71 | 6M to 3Y | 24 | >2Y | 55 | 6M to 5Y | 24 |
| **Older adults** | >60Y | 97 | >65Y | 65 | >65Y | 34 | >60Y | ND | >65Y | 32 |
| **Persons with chronic diseases** | Yes | ND | Yes | 100* | Yes | 100* | Yes | 99* | Yes | ND |
| **Pregnant women** | Yes | 81 | Yes | 90 | Yes | 28 | Yes | 38 | Yes | 25 |
| **Health personnel** | Yes | 95 | Yes | 100 | Yes | 54 | Yes | ND | Yes | 57 |
| **Others** | Teachers, indigenous people, prisoners | ND | Chicken and pig farmers | ND | Teachers, caregivers to institutionalized persons, journalists, prisoners | ND | Security & and military personnel, prisoners and institutionalized persons; indigenous peoples, residents of cold areas | ND | None | ND |

M = Months; Y = Years; ND = No data.

*2017 for Chile, 2016 for Peru and 2013 for Paraguay.

Sources: PAHO. Coverage of influenza vaccine 2015. http://ais.paho.org/imm/InfluenzaCoverageMap.asp. For risk group characteristics: Influenza vaccination documents (see Table 2).

Brazil and Chile have considerably higher uptake for all risk groups as against Paraguay and Uruguay, and also to a lesser extent in comparison to Peru (Table 5).

Focus group participants generally expressed trust in their health care providers, with exceptions in Paraguay. Even when they had doubts about the vaccine, their confidence in health professionals leads them to gain confidence in the vaccine. However, the medical advice was perceived as ambiguous across risk groups, expressing that sometimes they were instructed to be vaccinated, others not.

Our study also suggests that less information or knowledge is associated to a greater mistrust of the vaccine. Paraguay stands out as a country where focus group participants had very little knowledge of both the disease and the vaccine.

The importance of confidence as a factor in vaccine hesitancy was also observed by the survey arm of our study which obtained quantitative information on the 3Cs for the same risk groups, countries and socioeconomic groups, reporting that individuals with a higher composite indicator of influenza vaccine confidence also had higher probability of being vaccinated in the last year [15]. This study also suggested higher vaccine confidence and vaccine uptake levels in Chile and lower in Paraguay and Uruguay, supporting the qualitative study results we are now reporting.

Other studies in South America also evaluate confidence as the critical component of vaccine hesitancy. Brown et al. found among a survey sample in Brazil that the commonest reasons for hesitancy were issues with confidence (41.4%), efficacy/safety of the vaccine (25.5%) and concerns about adverse events (23.6%) [16]. Reinders and collaborators found in a randomized household survey in urban areas near large hospitals in Peru that confidence and specifically "being afraid of vaccination and its effects" is the most critical factor behind influenza vaccine hesitance. A world-wide systematic review of seasonal influenza vaccination intention and behavior undertaken by WHO between 2005 and 2016 found confidence and specifically

a perceived low effectiveness of the vaccine as the most frequently reported reasons for hesitancy across risk groups [16].

Complacency with not being vaccinated was found in our study to be also critical for vaccine acceptance, with a large and diversified set of perceptions such as the importance of a healthy diet lowering the perception of influenza risks while providing alternatives for prevention. We also found a low level of perception of at-risk individuals as members of a risk group, except for older adults and to some extent adults with risk factors, especially those who had a condition such as asthma. The tendency was to perceive greater risk in others, particularly among pregnant women who perceive vaccination to be more important for their babies than for themselves. Across countries, complacency of influenza was higher in Paraguay and Uruguay than in other countries, possibly due to the high value given in these countries to traditional measures to prevent illness, a situation that could take the vaccine out of the spotlight or relegate as a secondary preventive measure.

Our data indicate that confidence and complacency are intimately related. We found that complacency with the risks posed by influenza in the context of trust in traditional medicine and distrust in the health system–as observed for the case of Paraguay–was associated to a perception of side-effects and even of the view that the vaccine can cause influenza. Furthermore, our study suggests, as in the case of older adults, that low levels of complacency with influenza risks can support higher levels of confidence in the vaccine. These possible associations across confidence and complacency require further research to ascertain cognitive and affective links.

In our study, less information and knowledge regarding influenza and the vaccine are qualitatively related to a greater mistrust of vaccination. Focus group participants from Paraguay expressed the most doubts on both topics. The more people distrust the vaccine, the more they will have recourse to traditional or alternative strategies to prevent influenza, such as diet, hygiene, clothing, environment. Those who do not perceive influenza to be a serious or life-threatening disease tend to be more vaccine hesitant.

Findings on confidence and complacency point to the importance of health promotion and communication strategies to improve knowledge about influenza risks and to establish that hygiene measures do not replace vaccination. While all risk groups acknowledged that influenza could be a potentially mortal disease, they also expressed confusion between influenza and other respiratory infections. In this sense, influenza is seen either as a complication of the common cold or as something very prevalent–ordinary even–posing a small threat to the health of most people who have it.

Reinders and collaborators found particularly low levels of self-perception of pregnant women and older adults as high-risk groups for influenza in Peru [16]. The perception of individuals as belonging to risk groups can be enhanced through conditioning social program benefits, as in the case of children in Brazil, where having a vaccination card up to date is a requirement to receive Bolsa Família, a government program for income redistribution [18].

For the past decades, social scientists have discussed how different Explanatory Models (EM) for a disease are important for understanding how individuals explain their causes, symptoms, and treatments, and hence for the role of complacency in vaccine hesitancy. The flu and the common cold are associated by participants to feeling cold or exposing parts of the body to lower temperatures (bare feet specially), and prevention involves dressing warmly or consuming warm foods and liquids. A healthy diet is cited by participants as another important preventive measure, and that involves consuming fruits rich with vitamin C or associated with boosting the immune system. In our findings, influenza vaccination is perceived as competing with other practices that are also considered preventative, whether by themselves or as complements to vaccination. Mothers of young children and pregnant women tend to over rely on hygiene and a healthy diet–which are unquestionably positive for pregnancy and child

development–as preventative of influenza. For the older adults and adults with risk factors, their perceived vulnerability to influenza may be associated with the observed expressions of interest in vaccination, making them less complacent. Future studies focusing on folk medicine and existing EM for the flu and the cold in urban contexts of Latin America would be important to expand on our findings [19, 20].

Our study suggests that individuals belonging to specific risk groups for influenza lack examples of the severity and hence the uniqueness of influenza, possibly due to the paradoxical effects of vaccination programs as they reduce mortality. The exception were older adults, who had both more personal experience with the vaccine and knew someone who had been terribly ill due to influenza. Reinders and collaborators found greater vaccination rates among persons who perceived the severity of influenza in Peru [16]. Our survey study found complacency to be the component of the 3C model with the weakest evaluation given prejudices, knowledge and risk perception. However, as mentioned already, complacency was not as strongly associated with vaccination rates as confidence was. Further afield, in Australia extreme events lived by communities pivoted parental responsibility for child vaccination [21].

The convenience of influenza vaccination was the most positively evaluated component in our study, both across risk groups and countries. None-the-less, perceptions on restrictions with information, particularly in Brazil, were observed, while opportunities to improve access were found across all risk groups and countries. Perceptions of program appeal were not so much negative as absent, together with examples of negative appeal in terms of possible disrespectful and culturally inappropriate treatment of indigenous population in Peru, as found for health care in general for indigenous populations in other countries [22, 23]. In another arm of our research, we explored the balance given by country influenza vaccination programs to strategies focused on increasing confidence and convenience and on lowering complacency [24]. Congruent with the qualitative findings, we found that programs across the five countries privilege convenience through supply-side strategies. Our qualitative findings suggest the need for influenza vaccination programs to address confidence and complacency through directed strategies.

A limitation of our qualitative research is its primary focus on the situation of vaccine hesitancy within risk groups rather than at the country level. Furthermore, our results cannot be generalized to urban populations across countries, given the primary aim of the research of identifying qualitative patterns. Even so, our sample aimed to include countries with both high performing vaccine programs (Brazil and Chile) and low-performing programs (Paraguay, Peru and Uruguay, see Table 5). A possible limitation of the focus group technique employed is that it could have inhibited responses that differed markedly from those held by the majority of the participants.

In spite of the limitations stated, our analysis did suggest that influenza vaccination hesitancy has greater similarities than differences across the countries studied, possibly due to similar social, health system and cultural characteristics, as in our analysis of national vaccination programs [24].

The specific findings regarding each of the 3Cs in the vaccine hesitancy model across risk groups and countries provide valuable information for program planning and to guide intersectoral policies and strategies. Given that vaccine convenience perceptions are structured and focused on limited problems, efforts to improve this dimension of hesitancy can be readily implemented and can support strategies to improve confidence and complacency. Effective and assertive communication strategies can be developed to target specific risk groups and health workers involved in vaccination. Messages should emphasize the risks posed by influenza and the benefits of being vaccinated, and these messages need to be divulged all year, not only during the influenza vaccination campaigns. Spokespersons can be designated and

trained to demonstrate the value of vaccination as well as to solve all vaccination doubts through interactive, digital-based media as available. However, vaccine hesitancy is very often based on emotional responses and countering emotion with reason (e.g. unbiased information) typically doesn't work [24]. Bolstering vaccine confidence on the basis of emotive as well as rational concepts adapted to specific risk groups should therefore be the top priority to reduce hesitancy. Given that the most trusted sources of information are health professionals, efforts should be made to provide them with training, incentives and tools focusing on the needs and sensibilities of specific risk groups. Three groups of actions have been proposed: addressing opportunities to modify confidence in vaccine effectiveness and concerns about safety; bolstering altruism in specific social contexts through normative messaging; and pivoting change in health professionals through incentives, sanctions and requirements [25, 26]. Communication strategies must be precise to avoid counterproductive results which could lead to reinforcing hesitation to vaccination [27].

The fact that focus group participants across risk groups from Paraguay and Uruguay expressed greater influenza vaccine hesitancy suggest the importance of further analyzing country-specific factors such as the reliance on traditional medicine as well as differences in health system trust.

The COVID-19 pandemic will likely affect uptake for seasonal influenza vaccination given changes in risk perception associated with mortality and economic impact of viral respiratory diseases [28]. Influenza vaccination has been recommended to reduce hospitalizations and relieve resources to care for COVID-19. The COVID-19 vaccine may also alter seasonal influenza vaccination programs and influenza vaccine uptake. Seasonal influenza vaccination in Brazil was anticipated one month in March 2020 to help reduce respiratory diseases and help to cope with COVID-19. It is important to prepare for COVID-19 vaccination addressing lessons with influenza vaccine hesitancy. Influenza and COVID-19 vaccination strategies may converge in the future so our findings can be useful to prepare for such a scenario.

## Conclusions

The study of influenza vaccine hesitancy in five countries of South America revealed a wide range of perceptions regarding confidence and complacency across risk groups and, to a lesser extent, across countries. While risk groups share some common perceptions, they also have specific conceptions that present intervention opportunities. Information and communication strategies need to be tailored for specific risk groups to address their concerns and the take advantage of their varied relationship to health providers. Country-specific measures should consider contextual factors. Confidence and complacency should be addressed in their own right, through strategies that can increase one while reducing the other, in synergy.

## Supporting information

**S1 File. Focus group guide.**
(DOCX)

## Author Contributions

**Conceptualization:** Miguel Ángel González-Block, Blanca Estela Pelcastre-Villafuerte, Sandra Patricia Díaz Portillo, Elsa Sarti.

**Data curation:** Daniela Riva Knauth, Andréa Fachel-Leal, Yamila Comes, Pedro Crocco, Laura Noboa, Berenice Rodríguez Zea, Mónica Ruoti, Sandra Patricia Díaz Portillo.

**Formal analysis:** Miguel Ángel González-Block, Blanca Estela Pelcastre-Villafuerte, Andréa Fachel-Leal, Sandra Patricia Díaz Portillo.

**Funding acquisition:** Miguel Ángel González-Block, Elsa Sarti.

**Investigation:** Miguel Ángel González-Block, Daniela Riva Knauth, Andréa Fachel-Leal, Yamila Comes, Pedro Crocco, Laura Noboa, Berenice Rodríguez Zea, Mónica Ruoti.

**Methodology:** Miguel Ángel González-Block, Blanca Estela Pelcastre-Villafuerte.

**Project administration:** Miguel Ángel González-Block.

**Supervision:** Miguel Ángel González-Block.

**Validation:** Miguel Ángel González-Block.

**Writing – original draft:** Miguel Ángel González-Block, Blanca Estela Pelcastre-Villafuerte, Sandra Patricia Díaz Portillo.

**Writing – review & editing:** Miguel Ángel González-Block, Blanca Estela Pelcastre-Villafuerte, Daniela Riva Knauth, Andréa Fachel-Leal, Yamila Comes, Pedro Crocco, Laura Noboa, Berenice Rodríguez Zea, Mónica Ruoti, Sandra Patricia Díaz Portillo, Elsa Sarti.

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
