## [Decision Letter · Decision Letter 0]

26 Apr 2021

PONE-D-21-09160

Influenza vaccination hesitancy in large urban centers in South America. Qualitative analysis of confidence, complacency and convenience across risk groups

PLOS ONE

Dear Dr. González-Block,

Thank you for submitting your manuscript to PLOS ONE. After careful consideration, we feel that it has merit but does not fully meet PLOS ONE’s publication criteria as it currently stands. Therefore, we invite you to submit a revised version of the manuscript that addresses the points raised during the review process.

We look forward to receiving your revised manuscript.

Kind regards,

Prof. Anat Gesser-Edelsburg, Ph.D.

Academic Editor

PLOS ONE

Journal Requirements:

2. Thank you for stating in the text of your manuscript "consent was obtained from participants". In the ethics statement in the Methods and online submission information, please ensure that you have specified:

 - whether consent was informed

 - what type of consent you obtained (for instance, written or verbal, and if verbal, how it was documented and witnessed).

3. Please provide more information on the focus group structure and how data was collected from these in your methods section. In addition, if there was a semi-structured guide to topics covered, questions asked, themes explored, etc. please provide this as a supplementary file.

4a) If there are ethical or legal restrictions on sharing a de-identified data set, please explain them in detail (e.g., data contain potentially sensitive information, data are owned by a third-party organization, etc.) and who has imposed them (e.g., an ethics committee). Please also provide contact information for a data access committee, ethics committee, or other institutional body to which data requests may be sent.

4b) If there are no restrictions, please upload the minimal anonymized data set necessary to replicate your study findings as either Supporting Information files or to a stable, public repository and provide us with the relevant URLs, DOIs, or accession numbers. For a list of acceptable repositories, please see http://journals.plos.org/plosone/s/data-availability#loc-recommended-repositories.

5. Thank you for stating the following in the Financial Disclosure section:

This project was financially supported by Sanofi Pasteur. Elsa Sarti is an employee of Sanofi Pasteur and played a role in the design of the study. All other authors have no relevant conflicts of interest to report.

We note that you received funding from a commercial source: Sanofi Pasteur.

Reviewers' comments:

Reviewer's Responses to Questions

**Comments to the Author**

1. Is the manuscript technically sound, and do the data support the conclusions?

Reviewer #1: Partly

Reviewer #2: Partly

2. Has the statistical analysis been performed appropriately and rigorously? 

Reviewer #1: N/A

Reviewer #2: No

3. Have the authors made all data underlying the findings in their manuscript fully available?

Reviewer #1: No

Reviewer #2: Yes

4. Is the manuscript presented in an intelligible fashion and written in standard English?

Reviewer #1: Yes

Reviewer #2: Yes

5. Review Comments to the Author

Reviewer #1: This manuscript describes the finding of an impressively large focus group study on vaccine hesitancy with a total of 20 focus groups conducted in 5 South American countries. It's greatest strengths are the obvious timeliness of the topic and the richness of the data, which come through in the authors ample use of quotations from participants. The paper is written is clear, easy-to-follow prose without jargon, making it accessible to a broad audience (there are a fair number of typos, though, so additional proofreading is recommended).

Nevertheless, I have a number of fairly significant concerns about the manuscript in its current form, mainly as data analysis is concerned, but also with several other aspects.

1. Data analysis procedures are not described in sufficient detail. The authors merely specify that data were coded thematically and don't provide any details on how exactly themes were identified, how coding with multiple coders was coordinated and how consistency was ensured, etc. Significantly more is needed here.

2. The authors do not provide any explanation of why focus groups where chosen as opposed to, e.g., individual interviews. Especially with potential concerns about health information and social desirability bias, I'm not convinced this was the right choice -- it definitely requires a justification as part of the paper's methods section.

3. The paper doesn't provide sufficient information about data collection. It would be common to provide basic demographic information about FG participants. I'd also want to know a bit more about recruitment strategies as well as the design of the focus group discussions. Generally speaking, I would encourage authors to follow reporting guidelines for qualitative research such as COREQ (https://doi.org/10.1093/intqhc/mzm042 )

4. The paper's analysis remains somewhat superficial. It gives a point-by-point discussion of the different themes identified, but does little to relate them to each other or make sense of them in the context of the different participating groups. Tables 2-4 are, on the one hand, useful as summaries, but on the other so detailed as to hide any significant analytic points that could be learned from comparing across different risk groups. Given the significant effort involved in conducting a five-country study, it's also somewhat disappointing that the authors do not take any advantage of this in systematically comparing across cases.

5. The data availability statement is insufficient as per the PLOS data policy. If data can be shared, it should be deposited in a data repository, not per request from the lead author. If the ethics protocol does not allow data sharing (which would be understandable given the nature of the data), this needs to be clearly stated, though authors should consider at a minimum a detailed summary of themes and associated descriptions and excerpts in that case.

Taken together, these are significant shortcomings. If editors decide to invite the authors to submit a revised version (which may well be warranted in spite of these shortcomings, due to the richness of the underlying data), I would encourage significant revisions with a focus on methods and methodology.

Reviewer #2: 1. FGD process could have been documented better

2. Not clear - whether double note takers were there to cross check the tonality and non verbal cues of the participants

3. RICE analysis could be done on FGD audio transcript to measure frequency of usage of significant phrases

4. FGD could follow maximum variance sampling instead of homogenous grouping to capture diversity of views

5. It could analyze the inter regional variance in perception

6. Some kind of multidimensional perception mapping schematic could be presented from thematic analysis

6. PLOS authors have the option to publish the peer review history of their article (what does this mean?). If published, this will include your full peer review and any attached files.

Reviewer #1: **Yes: **Sebastian Karcher

Reviewer #2: **Yes: **Arindam Ray

---

## [Author Response · Author response to Decision Letter 0]

21 May 2021

Focus group methodology

Reviewer 1 

2. The authors do not provide any explanation of why focus groups where chosen as opposed to, e.g., individual interviews. Especially with potential concerns about health information and social desirability bias, I'm not convinced this was the right choice -- it definitely requires a justification as part of the paper's methods section.

RESPONSE: We now provide a more detailed justification for the selection of FG methodology. 

Reviewer 1 

3. The paper doesn't provide sufficient information about data collection. It would be common to provide basic demographic information about FG participants. I'd also want to know a bit more about recruitment strategies as well as the design of the focus group discussions. Generally speaking, I would encourage authors to follow reporting guidelines for qualitative research such as COREQ (https://doi.org/10.1093/intqhc/mzm042 )

RESPONSE: We added complementary information regarding recruitment strategies and focus group design. Table 1 was expanded to provide further information regarding the number of participants by sex and risk group, the recruitment strategies, and the average age and number of children in the case of the mothers' groups. 

Reviewer 2

1. FGD process could have been documented better

RESPONSE: Please see response to Reviewer 1.

Reviewer 2

4. FGD could follow maximum variance sampling instead of homogenous grouping to capture diversity of views

RESPONSE: We now further justify how and why we recruited focus group participants based on homogeneous as well as heterogeneous risk-group characteristics, this last related to vaccination status. Table 1 provides more detailed information on focus groups composition. 

Reviewer 2

2. Not clear - whether double note takers were there to cross check the tonality and non verbal cues of the participants

RESPONSE: We did not use this focus group technique as we did not consider it relevant to code and analyse tonality and non-verbal cues.

Data analysis procedures

Reviewer 1 

1. Data analysis procedures are not described in sufficient detail. The authors merely specify that data were coded thematically and don't provide any details on how exactly themes were identified, how coding with multiple coders was coordinated and how consistency was ensured, etc. Significantly more is needed here.

RESPONSE: Greater detail is now provided in the methods section regarding the methods used for the development of codes and data interpretation in relation to the 3C conceptual framework. 

Reviewer 1 

4.1 The paper's analysis remains somewhat superficial. It gives a point-by-point discussion of the different themes identified, but does little to relate them to each other or make sense of them in the context of the different participating groups. Tables 2-4 are, on the one hand, useful as summaries, but on the other so detailed as to hide any significant analytic points that could be learned from comparing across different risk groups.

RESPONSE: The results section compares findings across risk groups in what we believe is appropriate detail. Eg., in the case of safety, we state that "All groups expressed... that the vaccine is safe ...except pregnant women, informants of all risk groups expressed concerns." In the case of trust in the health system, "Confidence in the information offered by service providers varies across risk groups". However, we have now complemented the analysis highlighting similarities and differences across countries, as requested. Furthermore, in the Discussion section we provide further detail on how the various themes relate across each other, highlighting their implications for vaccine hesitancy. We now provide Figure 1 highlighting key findings along the critical dimensions of confidence and complacency.

Reviewer 1 

4.2 Given the significant effort involved in conducting a five-country study, it's also somewhat disappointing that the authors do not take any advantage of this in systematically comparing across cases.

RESPONSE: The data-bases were reanalyzed in order to identify significant differences between countries. At the end of each results subsections a paragraph was included highlighting the most relevant findings considering similarities and differences between countries in the item. Main differences found across countries were addressed in the discussion section and the conclusions were modified accordingly. 

Reviewer 2

5. It could analyze the inter regional variance in perception

RESPONSE: Please see response to Reviewer 1.

Reviewer 2

3. RICE analysis could be done on FGD audio transcript to measure frequency of usage of significant phrases

RESPONSE: We appreciate the potential of the RICE analysis suggested, but this is outside the scope of the paper. Indeed, our aim as to describe and analyse participants perceptions of vaccine hesitancy, rather than the frequency of such perceptions.

Reviewer 1 

5.2 Authors should consider at a minimum a detailed summary of themes and associated descriptions and excerpts in that case

RESPONSE: Tables 2-4 aim to provide a detailed summary of themes. As stated in our response to point 4.1 above, we now provide Figure 1 with a caption that further summarizes key findings and dimensions. 

Reviewer 2

6. Some kind of multidimensional perception mapping schematic could be presented from thematic analysis

RESPONSE: The schematic is now provided in Figure 1.

Data availability

Reviewer 1 

5.1 The data availability statement is insufficient as per the PLOS data policy. If data can be shared, it should be deposited in a data repository, not per request from the lead author. If the ethics protocol does not allow data sharing (which would be understandable given the nature of the data), this needs to be clearly stated-

RESPONSE: Data is now being uploaded to a data repository. The data availability statement will show which repository we use.

---

## [Decision Letter · Decision Letter 1]

14 Jun 2021

PONE-D-21-09160R1

Influenza vaccination hesitancy in large urban centers in South America. Qualitative analysis of confidence, complacency and convenience across risk groups

PLOS ONE

Dear Dr. González-Block,

Thank you for submitting your manuscript to PLOS ONE. After careful consideration, we feel that it has merit but does not fully meet PLOS ONE’s publication criteria as it currently stands. Therefore, we invite you to submit a revised version of the manuscript that addresses the points raised during the review process.

We look forward to receiving your revised manuscript.

Kind regards,

Prof. Anat Gesser-Edelsburg, Ph.D.

Academic Editor

PLOS ONE

Journal Requirements:

Reviewers' comments:

Reviewer's Responses to Questions

**Comments to the Author**

1. If the authors have adequately addressed your comments raised in a previous round of review and you feel that this manuscript is now acceptable for publication, you may indicate that here to bypass the “Comments to the Author” section, enter your conflict of interest statement in the “Confidential to Editor” section, and submit your "Accept" recommendation.

Reviewer #1: All comments have been addressed

2. Is the manuscript technically sound, and do the data support the conclusions?

Reviewer #1: Yes

3. Has the statistical analysis been performed appropriately and rigorously? 

Reviewer #1: N/A

4. Have the authors made all data underlying the findings in their manuscript fully available?

Reviewer #1: Yes

5. Is the manuscript presented in an intelligible fashion and written in standard English?

Reviewer #1: Yes

6. Review Comments to the Author

Reviewer #1: Thank you for the revisions. Overall the manuscript is significantly improved and I only have relatively minor remaining comments. Specifically I think my comments about cross-country comparison and description of FG methodology have been fully addressed.

Some areas should still be improved before publication, though:

1. I think the justification for focus groups remains weak. A 1993 book saying the technique has been "validated" isn't a helpful citation: The concern isn't that focus groups aren't a valid form of qualitative inquiry (they are!), but whether they are the best form for the specific question at hand and what trade off their use has compared to alternative methods (different forms of semi-structured interviews, e.g.). Authors may decided that this works better in the discussion/limitations section of their paper rather than their methods section (this is just a suggestion -- as long as it is explicitly discussed, I have no preference as to where). For example, authors allude to "participant interaction" in focus groups -- a major strength of the method. It's unclear, however, why interaction is important for the individual-level attitudes authors are interested in. E.g., none of the presented excerpts show elements of interaction. I'm not saying it isn't important, but it is incumbent on the authors to make that case.

2. While the discussion of the coding methodology is significantly improved, I'm still not sure how consistency of coding was ensured. Multiple people coded the transcripts -- but each transcript was coded by only one author? By multiple coders in a group? After coding, author BP checked every code? Did they then overrule any disagreements? Were there none? The coding process a crucial part of qualitative methodology. Readers need to be able to understand fully and in detail how it was conducted. If there are elements that would be tedious or overly lengthy in the main article, pointing to a supplementary file is encouraged.

3. A minor point: Now that authors do much more with the cross-country comparison (which, to re-iterate, I think is great), I'd appreciate a small table comparing the countries, in particular in terms of vaccination rates (it's up to the authors if they want to include any other country characteristics).

7. PLOS authors have the option to publish the peer review history of their article (what does this mean?). If published, this will include your full peer review and any attached files.

Reviewer #1: **Yes: **Sebastian Karcher

---

## [Author Response · Author response to Decision Letter 1]

9 Jul 2021

• We further justify the use of focus group methodology as against alternatives, considering the advantages and limitations

• In the discussion section we address possible limitations of the focus group technique

• We have further detailed the coding methodology

• We added a table providing comparative data on influenza immunization rates by risk group across the five study countries and complemented our discussion with reference to said table

---

## [Editor Report · Decision Letter 2]

29 Jul 2021

Influenza vaccination hesitancy in large urban centers in South America. Qualitative analysis of confidence, complacency and convenience across risk groups

PONE-D-21-09160R2

Dear Dr. González-Block,

We’re pleased to inform you that your manuscript has been judged scientifically suitable for publication and will be formally accepted for publication once it meets all outstanding technical requirements.

Kind regards,

Prof. Anat Gesser-Edelsburg, Ph.D.

Academic Editor

PLOS ONE
---

## [Editor Report · Acceptance letter]

4 Aug 2021

PONE-D-21-09160R2 

Influenza vaccination hesitancy in large urban centers in South America. Qualitative analysis of confidence, complacency and convenience across risk groups 

Dear Dr. González-Block:

I'm pleased to inform you that your manuscript has been deemed suitable for publication in PLOS ONE. Congratulations! Your manuscript is now with our production department. 

Kind regards, 

on behalf of

Prof. Anat Gesser-Edelsburg 

Academic Editor

PLOS ONE